# Implementation of a care-pathway at the emergency department for older people presenting with nonspecific complaints; a protocol for a multicenter parallel cohort study

**M. G. A. M. van der Velde**[1,2]*, **M. A. C. Jansen**[3], **M. A. C. de Jongh**[3], **M. N. T. Kremers**[1,4], **H. R. Haak**[2,5]

**1** Department of Internal Medicine, Máxima MC, Veldhoven, The Netherlands, **2** Department of Health Services Research, and CAPHRI School for Public Health and Primary Care, Aging and Long Term Care Maastricht, Maastricht, The Netherlands, **3** Netwerk Acute Zorg Brabant, Tilburg, The Netherlands, **4** Department of Internal Medicine, Catharina Hospital, Eindhoven, The Netherlands, **5** Department of Internal Medicine, Maastricht University Medical Center, Maastricht, The Netherlands

* marleen.van.der.velde@mmc.nl

## Abstract

### Background

Older adults frequently attend the Emergency Department (ED) with poorly defined symptoms, often called nonspecific complaints (NSC). NSC such as 'weakness' and 'not feeling well', often lead to an extensive differential diagnosis. Patients with NSC experience a prolonged length of stay at the ED and are prone to adverse outcomes. Currently, a care-pathway for patients with NSC does not exist. A special structured care pathway for patients with NSC was designed to improve the efficiency and quality of care at the ED.

### Method

A multicenter parallel cohort study, organized in different hospitals in the Noord-Brabant area, the Netherlands, in which general practitioners (GP), elderly care physicians (ECP), Emergency Physicians (EP), geriatricians and internists will collaborate. Patients $\geq$ 70 years presenting with NSC and in need of ED admission as indicated by their own GP or ECP are eligible for inclusion. Before implementation each hospital will retrospectively include their own control-group. After implementation, patients will prospectively be included. The care-pathway exists of risk stratification by the APOP-screener, in-depth history taking, i.e. limited comprehensive geriatric assessment (CGA) and a standard set of diagnostics, and a dedicated ED-nurse (if possible) present to ensure the care-pathway is followed. The primary outcome is length of stay at the ED (LOS-ED) and perceived quality of care. Secondary outcomes are hospital length of stay, revisits, readmissions and mortality at 30- and 90-day follow-up.

**Data Availability Statement:** This paper does not report any data and the data availability policy is not applicable to this article.

**Funding:** The authors received no specific funding for this work.

**Competing interests:** The authors have declared that no competing interests exist.

## Discussion

This study proposes a structured care pathway for older patients presenting at the ED with NSCs and considering effectiveness and perceived quality this may improve acute care for these patients.

## Trial registration

Dutch Trial register, number NL8960.

## Background

Older adults frequently attend the Emergency Department (ED) with poorly defined symptoms, often called nonspecific complaints (NSC) [1, 2]. The concept of NSC is defined as all complaints that are not part of the set of specific complaints or signs, or where an initial working diagnosis cannot be definitely established [3]. NSC, such as 'not feeling well', 'being tired', unexplainable falling or simply being unable to cope with usual daily activities lead to a variety of differential diagnosis, ranging from social problems to life threatening conditions [3, 4]. Furthermore, 50% of the patients experience an acute medical problem at the time of presentation with a NSC [3, 5].

Studies have shown that patients with NSC have different characteristics compared to patients with specific complaints (SC) [6]. Patients with NSC have more comorbidities and polypharmacy than patients with SC. These factors combined with problems in functional status or communication lead to an increased complexity of the diagnostic process and under-triage, resulting in a high proportion of incorrect diagnoses and a great variety of discharge diagnosis in this population [3, 7, 8]. Consequently, patients with NSC are more frequently discharged with unrecognized and untreated health problems, have a longer length of stay at the ED (LOS-ED), higher admission rates and are more prone to adverse outcomes, such as mortality [9–12].

Treating older patients with NSC is challenging, due to the complexity of NSC and lack of a structured protocol evaluating these complaints. Several interventions in organization of care for this group have been tested, for example geriatric screening at the ED, geriatric EDs or passing the ED straight to the geriatric ward, but identifying patients at the highest risk is challenging and therefore rarely used in practice [13]. The development of the Acutely Presenting Older Patient (APOP)-screener, provides a way to identify older patients at the ED with a high risk of short and long-term poor outcomes[13, 14]. However, risk stratification should be the first step and a structured approach in managing patients with NSC is needed. Therefore, we aim to implement an integrated care pathway which will structure and streamline the acute care for patients with NSC from arrival at the ED until discharge.

We hypothesize that the NSC-care pathway will primarily improve patient satisfaction and reduce the LOS-ED. Reduced hospital length of stay, decreased 30-day mortality, a reduction of (re-)admission rates for older adults with NSC and reduced costs of care for this patient population might follow as a result of implementing this care pathway. Our study can be used to guide a management protocol for older patients with NSC at the ED to further improve the quality of care.

## Methods

### Objective

The aim of this study is to implement and evaluate a care pathway for older adults presenting at the ED with NSC. The secondary aim of our study is to evaluate the standardization and need of the included diagnostic tests in the care-pathway.

### Trial design

This study is a longitudinal multicenter parallel cohort trial. The study will be organized in hospitals in the Noord-Brabant area, the Netherlands, in which general practitioners (GPs), elderly care physicians (ECPs), Emergency Physicians (EPs), geriatricians and internists will collaborate. Participating hospitals can be general or teaching hospitals.

### Eligibility criteria

Older patients ($\geq$ 70 years) with NSC referred to the ED by the GP or ECP will be screened for inclusion. Patients who are referred by their known GP or ECP within working-hours (8:00–17:00 on weekdays) are eligible, due to the accessibility of the patients personal GP-office to retrieve further medical or social data when needed and the different organization of acute care out-of-hours [15]. Subsequent presentation at the ED is possible between 8:00 and 20:00, taking logistic difficulties as transport into account. Patients can also enter the care pathway if triage at the ED indicates NSC. In order to be eligible to participate in this study, patients have to meet the following criteria; 1) indicated for admission to the ED, and 2) age $\geq$ 70 years, and 3) a nonspecific complaint at presentation. NSC are divided into five referral categories: 1) somatic problems, such as weakness, not feeling well, change in nutritional status or unexplained weight loss; 2) an increased demand of care, such as loss of independency, a necessity for change in the living situation or 24–7 care, not indicated previously; 3) cognitive problems, such as disorientation, changes in behavior, abrupt cognitive decline; 4) a decline in functional status, such as loss of mobility; and 5) unexplained falls, not related to extrinsic factors.

### Patient recruitment, randomization and collection of data

This study is a parallel cohort study with a baseline period. Before implementing the care pathway each hospital will include their own control group using a graphic user interface data mining tool with text-mining features (CTcue, version 3.0; Amsterdam, the Netherlands) [16]. Ctcue is a search engine in which unstructured data from an Electronic Health Record (EHR) can be easily found using specific queries. Data will be retrieved according to our defined in- and exclusion criteria, study parameters and endpoints anonymously, therefore a waiver for informed consent has been obtained. When Ctcue is not available, the control group will be included prospectively. The ED physician will save the data of eligible patients in the care pathway (before it is implemented). The specialist in charge of the patient is allowed to use patient data for research without consent, when data is pseudonymized. The specialist may grant this right to a third party, in this case the research team. Exchange of data has to be approved by the involved departments, i.e. Internal Medicine, Geriatrics and the Emergency Department (Fig 1). Given the unprecedented strain on acute care services, we decided not to perform a randomized trial as we estimated this as non-feasible. Therefore, we start the intervention period conform convenience, sequentially following the capacity of participating hospitals to implement the pathway (Fig 2). Hence, we will not include a transition period [17]. Control patients will be included retrospectively after the start date of implementation, to prevent information bias.

| | STUDY PERIOD | | | | |
|---|---|---|---|---|---|
| | Enrolment | Allocation | Post-allocation | | Close-out |
| **TIMEPOINT\*\*** | *-t₁* | **0** | $t_1$<br>*(30 days after ED visit)* | $t_2$ *(90 days after ED visit)* | *End of follow-up ($T^{91}$)* |
| **ENROLMENT:** | | | | | |
| **Inclusion control group** | 4 months prior to study | | | | |
| **Eligibility screen** | X<br>(At referral/arrival at the ED) | | | | |
| **Informed consent** | | X | | | |
| **Allocation** | | X | | | |
| **INTERVENTIONS:** | | | | | |
| *[Care-pathway NSC at the ED]* | | | ●━━━━━━━━━━━● | | |
| **ASSESSMENTS:** | | | | | |
| *[Baseline variable assessment]* | X | X | | | |
| *[Outcome variables]* | | | X | X | X |

**Fig 1. Schedule of enrolment, interventions, and assessments.**

Intervention patients will receive written patient information and will subsequently be asked for informed consent at the end of the ED-visit. If a patient has impaired cognitive function at the time of presentation at the ED, the caregiver or relative may grant their permission for inclusion. Patients are asked to fill out a questionnaire regarding patient satisfaction, when given informed consent. Patient satisfaction will be measured by the PRM-acute care (S1 File) [18]. This is a valid instrument to measure the perceived quality of healthcare in an acute setting in five domains: relief of symptoms, understanding the diagnosis, presence and understanding of the diagnostic and/or therapeutic plan, reassurance and patient experiences. We

| | | | | | | | | | | | | |
|---|---|---|---|---|---|---|---|---|---|---|---|---|
| Hospital 1 | | | | | | | | | | | | |
| Hospital 2 | NA | | | | | | | | | | | |
| Hospital 3 | NA | NA | | | | | | | | | | |
| Hospital 4 | NA | NA | NA | | | | | | | | | |
| Etc. | NA | NA | NA | NA | | | | | | | | |

**Fig 2. A parallel cohort study with baseline period.** NA: Not applicable. Light blue: Control. Dark blue: Intervention.

added two questions to the validated questionnaire, for further evaluation of the care pathway. These questions involve the patient experience of their complaints being treated (question 3) and the experience of needing further examination (question 8). When a patient is unable to fill out the questionnaire, for instance due to impaired cognitive function, he/she can still participate in the study but the patient questionnaire cannot be used and will be considered missing data. Data of ED-revisits, readmissions and mortality will be gathered at 30 and 90 days after the ED-visit.

## Ethics

The study will be conducted according to the principles of the Declaration of Helsinki (version 2004, May 22, 2007; www.wma.net) in accordance with the Medical Research Involving Human Subjects Act (WMO). This study is approved by the ethics committee at the Máxima Medical Centre in Veldhoven (ref. no N19.034) and by participating hospitals. The ethics committee judged the pathway as standard care, therefore patients eligible for the pathway will be included and are asked for informed consent at the end of the ED-visit. Data handling will be done pseudonymized. Informed consent forms will be stored locally. Data will be collected in a digital research platform (Research Manager, version 6.8, Deventer, The Netherlands), and in accordance with guidelines of the Dutch Federation of University Medical Centers (NFU) the data will be kept for 20 years [19]. The protocol is registered in the Dutch Trial Register, number NL8960. The authors received no specific funding for this word and declared that no conflict of interest exists. We expect to publish the results of the care pathway in a peer reviewed journal.

## Intervention

The structure and content of the care pathway has been established by review of literature regarding NSC at the ED, followed by a total of eight expert meetings with input from physicians, (ED-) managers, counsellors acute care and epidemiologists from different departments (i.e. emergency, internal, geriatric and general medicine).

The care pathway will comprise of special attention and focused care for the patient with NSC at the ED (Fig 3). All older adult patients presenting with a nonspecific complaint, can be referred to the NSC-care pathway at the ED by the GP or ECP. Upon referral to the ED, the GP or ECP and ED-physician review the name and address details for the patient, discuss advance care planning and review the reason for access to the NSC-care pathway following the five categories of NSC. Patients can also enter the care pathway if triage at the ED indicates NSC.

Upon arrival at the ED, patients will be triaged, including execution of the APOP-screener, which is integrated in the electronic health record. The APOP-screener is a validated

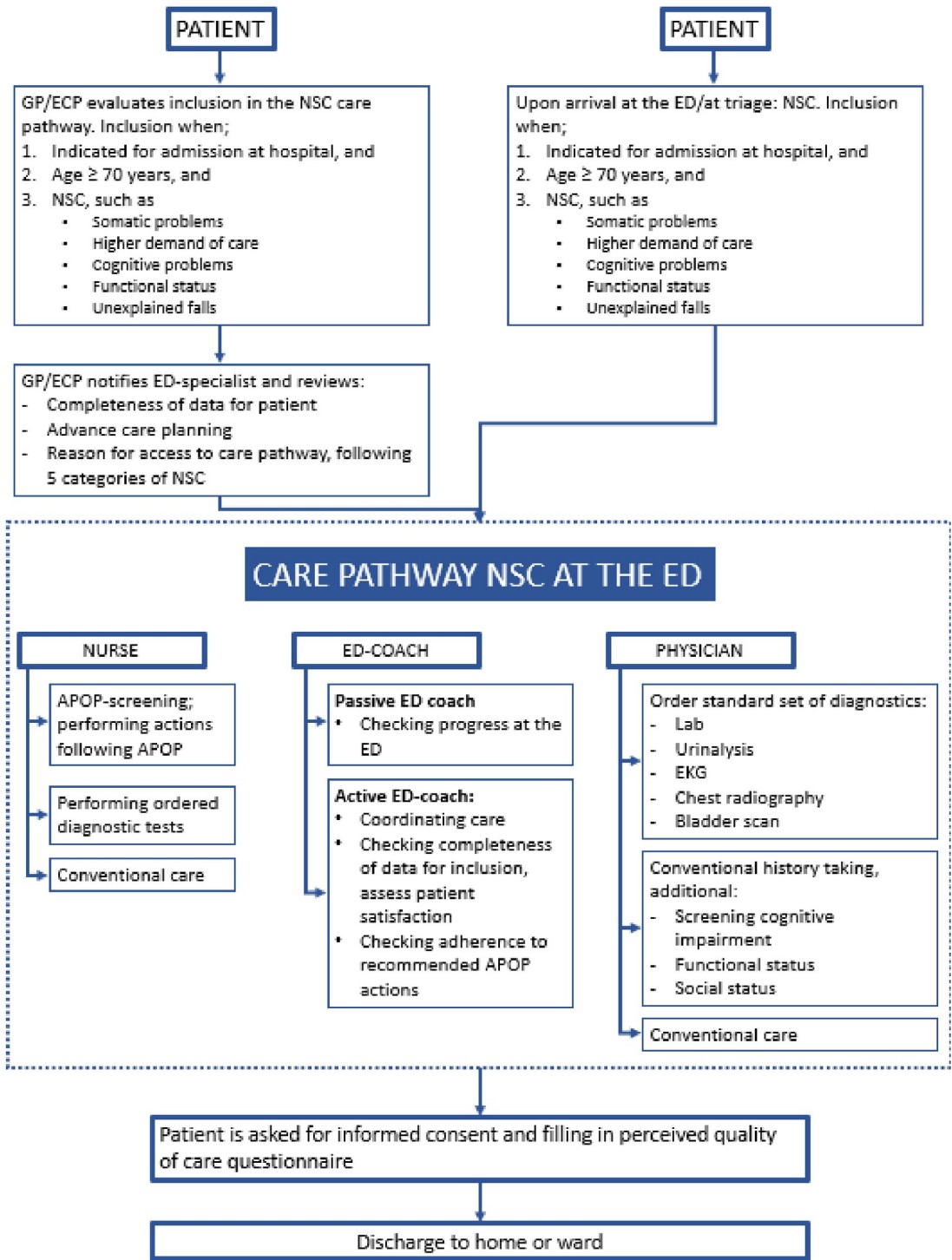

**Fig 3. Schematic overview of the care-pathway nonspecific complaints (NSC).** GP: General practitioner, ECP: Elderly care physician, ED: Emergency Department, APOP: Acutely Presenting Older Patient.

instrument to estimate frailty for older patients ($\geq$ 70 years old) presenting to the ED, expressed as risk for functional decline and mortality within three months and estimated cognitive impairment [13, 14]. This short questionnaire consists of questions regarding

performance status, medical history, gender and age, and can be completed in only 2 minutes. The estimated risk of frailty is expressed in four outcomes: low risk, high risk at functional decline, evidence of cognitive impairment and both risk of functional decline and evidence of cognitive impairment. When a patient is admitted at the hospital and scored unfavorable in the APOP screener, a consultation for a comprehensive geriatric assessment (CGA) lead by the internist geriatric medicine or clinical geriatrician with a team of allied health care professionals, should be considered

Additional to the APOP-screener and somatic history taking, treating physicians are asked to screen various domains following the CGA, i.e. cognitive, functional and social. The estimated extra time needed for performing the APOP screener and the 'acute' geriatric assessment is calculated at fifteen minutes per patient. Patients will be treated by an experienced doctor, according to the hospital policy. Medical students are excluded from participation in the pathway.

Due to the broad differential diagnosis and potential underlying diseases in patients presenting with NSC, we standardized the diagnostics for patients in the care pathway. These diagnostic measurements consists of an electrocardiogram (EKG), urinalysis, chest radiography (X-thorax), bladder scan and comprehensive bloodwork including complete blood count, electrolytes (sodium, potassium, calcium), kidney function, thyroid gland function and inflammation parameters. Further diagnostics measurements can be performed according to the physicians expertise. Medication verification will be done at the ED by the pharmacists assistants or the treating physician at the ED.

After establishment of the treatment plan, patients will be discharged from the ED and either be admitted to the ward, discharged home, to a care home, a short stay facility for primary care or a geriatric rehabilitation. If possible, a dedicated ED-nurse can be assigned to each patient in the care pathway to oversee the care process, check the progress at the ED and assess the participants satisfaction of the care pathway.

## Outcome measures

The primary outcome is LOS- ED and perceived quality of care on five domains. In the Netherlands the LOS-ED does not end when diagnosis is established and patients is referred to a specialty, it ends when the patient is discharged to the ward. LOS-ED is therefore associated with the number of consultations during the ED-visit, a higher number of diagnostic tests and experience of the evaluating physician [20]. By standardizing diagnostic tests and by agreeing experienced physicians will evaluate these patients, we expect to reduce the LOS-ED. To evaluate the effectiveness of the care pathway, secondary outcomes are hospital length of stay, discharge destination, frequency of revisits/readmissions, 30 and 90-day mortality, medical diagnosis (at admission versus discharge) and outcomes/appropriateness of the performed diagnostic measurements.

## Statistical analysis

**Sample size.** We hypothesize that implementing the care pathway will reduce the mean LOS-ED with 10%. In a prospective study about the quality of acute care by M.N.T. Kremers, preliminary data showed a mean LOS-ED in the Máxima Medical Centre of three hours and twenty minutes (95% CI 3:00–3:40, SD = 1.17) for patients presenting to the ED for internal medicine between January 5 and March 12, 2020 [21]. Assuming a normal distribution, a two-sided test, power of 80% and a significance level ($\alpha$) of 0.05, a minimum of 466 patients need to be included, 233 patients in both control and intervention group. We aim to include 300 patients in each group, divided over three participating hospitals, to guarantee power in case

of missing data. Each hospital will gather their own control-group previous to implementation of the care-pathway. A minimum number of control- and intervention patients will be established for each participating hospital, dependent of number of annual patient-visits at the Emergency Department. Control and intervention patients will be gathered following a 1:1 ratio per hospital.

### Data-analysis

All statistical analyses will be performed with SPSS 24.0 for Windows (IBM, SPSS Statistics, Chicago, IL, USA). Descriptive statistics will be used for frequencies, percentages, means and standard deviations variables with normal distribution and median and interquartile ranges for variables with non-normal distribution. Prevalence of the most frequent diagnosis (coded ICD-10) at discharge will be presented in percentages. Continuous variables will be analyzed using independent t-tests, whereas the categorical data will be analyzed by chi-square or Fisher's exact test. Non-normally distributed categorical and metric variables of the study will be analyzed using Mann-Whitney-U-test. All tests will be performed using a significance level of alpha 0.05. A two-sided p-value $<0.05$ will be considered significant.

### Timeline

Control patients were gathered within CTcue before implementation was started. This period covers patients presenting to the ED with NSC between December 2020 and date of implementation of the care-pathway. The first hospital implementing the care pathway started in April 2020. Currently, all participating hospitals have gathered control patients and have implemented the care-pathway in the ED. Patient inclusion is expected to be completed by December 2023. The data analysis will be conducted in 2024, and the manuscript will be completed at the end of 2024.

## Discussion

In view of the 'ever-increasing' older population presenting to the ED with NSC, optimization of acute care for this population is of upmost importance. Every patient deserves a goal-oriented and person centered approach, however due to the complexity of NSC and lack of a structured protocol evaluating these complaints, managing patients with NSC is challenging and they subsequently experience a higher risk of adverse health outcomes. This study proposes a structured care pathway for patients presenting at the ED with NSC. The care-pathway consists of risk stratification by the APOP-screener, in-depth history taking with screening the domains following the CGA and a standard set of diagnostics, with an active ED-coach (if possible) present to ensure the care-pathway is followed. The goal of our study is to evaluate the proposed care-pathway and the standardization and need of the included diagnostic tests in the care-pathway. The results of this study will help us to establish a management protocol for patients with NSC and to make further recommendations, for example tailormade diagnostic trajectories following referral category. We hypothesize this will improve efficacy and quality of care at the ED, and reduced costs of care might follow as a result.

- Mw. C. Schepel, director, Netwerk Acute Zorg Brabant, Tilburg

- Mw E.T.J. du Cloo, counselor acute care, Netwerk Acute Zorg Brabant, Tilburg

- Mw. Dr. K. Holtkamp, counselor acute care, Netwerk Acute Zorg Brabant, Tilburg

- Mw. C. de Vries, project-leader, Zorggroep DOH, Eindhoven

- Dhr. Drs. P. Wouda †, general practitioner, Huisartsencentrum Parklaan-Wouda, Eindhoven

- Mw. Drs. S.L.E Lambooij, Department of Internal Medicine, Máxima MC, Eindhoven/ Veldhoven

- Mw. Drs. M. Hermans, elderly care physician, de Wever, Tilburg

- Dhr. Drs. S. Elbouazati, Emergency Department, Bravis Hospital, Bergen op Zoom/ Roosendaal

- Mw. Drs. F. Horsten, Emergency Department, Bravis Hospital, Bergen op Zoom/ Roosendaal

- Mw. Dr. G. Buunk, Department of Internal Medicine, Amphia Hospital, Breda

- Dhr. Drs. C.A.S. Berende, Emergency Department, Amphia Hospital, Breda

- Mw. Dr. M. van der Velde, Department of Internal Medicine, Elisabeth Twee Steden Hospital, Tilburg

- Mw. Dr. C.J.P.W. Keijsers, Department of Geriatrics, Jeroen Bosch Hospital, 's Hertogenbosch

- Dhr. J. de Laat, Emergency Department, Jeroen Bosch Hospital, 's Hertogenbosch

- Dhr. Drs. R.W. Vingerhoets, Department of Geriatrics, Elisabeth Twee Steden Hospital, Tilburg

## Supporting information

**S1 File. PRM acute care.**
(DOCX)

**S2 File. SPIRIT 2013 checklist.**
(DOCX)

**S3 File.**
(PDF)

## Acknowledgments

We thank the following persons for their input and expertise in formulating this protocol:

## Author Contributions

**Conceptualization:** M. A. C. de Jongh, H. R. Haak.

**Data curation:** M. G. A. M. van der Velde.

**Formal analysis:** M. G. A. M. van der Velde, M. N. T. Kremers.

**Investigation:** H. R. Haak.

**Methodology:** M. G. A. M. van der Velde, M. A. C. Jansen, M. A. C. de Jongh, M. N. T. Kremers, H. R. Haak.

**Project administration:** M. G. A. M. van der Velde, M. A. C. Jansen.

**Supervision:** M. A. C. Jansen, M. N. T. Kremers, H. R. Haak.

**Writing – original draft:** M. G. A. M. van der Velde.

**Writing – review & editing:** M. A. C. Jansen, M. A. C. de Jongh, M. N. T. Kremers, H. R. Haak.

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
