## [Decision Letter · Decision Letter 0]

9 Jul 2023

PONE-D-23-17637Implementation of a care-pathway at the emergency department for older people presenting with nonspecific complaints; A protocol for a multicenter stepped wedged cohort studyPLOS ONE

Dear Dr. van der Velde,

Thank you for submitting your manuscript to PLOS ONE. After careful consideration, we feel that it has merit but does not fully meet PLOS ONE’s publication criteria as it currently stands. Therefore, we invite you to submit a revised version of the manuscript that addresses the points raised during the review process.

We look forward to receiving your revised manuscript.

Kind regards,

Antony Bayer

Academic Editor

PLOS ONE

a) If there are ethical or legal restrictions on sharing a de-identified data set, please explain them in detail (e.g., data contain potentially sensitive information, data are owned by a third-party organization, etc.) and who has imposed them (e.g., an ethics committee). Please also provide contact information for a data access committee, ethics committee, or other institutional body to which data requests may be sent. Please note that authors, including Corresponding Authors, are not permitted to be the sole point of contact for data requests.

b) If there are no restrictions, please provide the minimal anonymized data set necessary to replicate your study findings as either Supporting Information files or to a stable, public repository and provide us with the relevant URLs, DOIs, or accession numbers. For a list of acceptable repositories, please see http://journals.plos.org/plosone/s/data-availability#loc-recommended-repositories.

3. One of the noted authors is a group or consortium [The Orca Onderzoeks Consortium Acute Geneeskunde Acute Medicine Research Consortium HR]. In addition to naming the author group, please list the individual authors and affiliations within this group in the acknowledgments section of your manuscript. Please also indicate clearly a lead author for this group along with a contact email address.

Reviewers' comments:

Reviewer's Responses to Questions

**Comments to the Author**

1. Does the manuscript provide a valid rationale for the proposed study, with clearly identified and justified research questions?

Reviewer #1: Yes

Reviewer #2: Yes

2. Is the protocol technically sound and planned in a manner that will lead to a meaningful outcome and allow testing the stated hypotheses?

Reviewer #1: Yes

Reviewer #2: Yes

3. Is the methodology feasible and described in sufficient detail to allow the work to be replicable?

Reviewer #1: Yes

Reviewer #2: Yes

4. Have the authors described where all data underlying the findings will be made available when the study is complete?

Reviewer #1: Yes

Reviewer #2: Yes

5. Is the manuscript presented in an intelligible fashion and written in standard English?

Reviewer #1: Yes

Reviewer #2: Yes

6. Review Comments to the Author

You may also provide optional suggestions and comments to authors that they might find helpful in planning their study.

Reviewer #1: Nice proposal and a much needed study. I was unclear as to whether or not this was a cluster deisgn, and if so why no randomisation, and if you needed to adjust the sample size for the cluster design. Also the intervention made no mention of other members of the team required to deliver CGA such as therapists. Good to see a PROM in the outcomes, but perhaps also add in function?

Reviewer #2: Thank you for inviting me to review this manuscript presenting a protocol for a stepped wedged cohort implementation study.

Non-specific complaints are beginning to be recognised as the deadliest of presenting symptoms to geriatric emergency care. I might describe an NSC as a vaguely characterised symptom set outside of traditional teaching’s disease-based medical classification, which naively assumes that people can usually localise their problem. The classification presented in the methods is very helpful.

For an international audience, I wonder if you might consider changing your first sentence from “older adults are frequently referred to the ED” to “older adults frequently attend the ED…” – outside of Belgium and the Netherlands people more commonly attend without referral, and NSC are no less serious in these cases.

I notice the authors use a term “elderly care physicians”. I am interested to know how these professionals differ from geriatricians. In the UK, “elderly” has become an ageist slur and is a word best avoided if at all possible.

It is pleasing to see a patient-reported outcome being collected and that the researchers will include people with impaired cognition but who are able to answer the items. I wonder if you might consider adding a short statement acknowledging that capacity can fluctuate and that the participant will receive opportunity to re-attempt the PRM.

Regarding the care pathway, I am not sure I understand why the diagnostics are being standardised. Does this imply that every person attending with a NSC will receive every stated test, including the controversial urinalysis? This approach would contradict my understanding of frailty risk and NSC as prompts for goal-oriented, person-centred care in which an investigation plan is tailored to the individual. I can see this ‘shotgun approach’ leading to confirmation bias in clinical decision-making. I also cannot see how analysis will differentiate the value of investigations, as for some NSCs a test e.g. chest xray will be appropriate and useful, whereas for others it will not add value.

The study is being powered to observe for reduction in LOS-ED. This for me prompts interesting reflection – in my practice, the LOS-ED is due to hospital capacity and crowding rather than patient factors, but presumably in all the research settings here the LOS-ED ends when an initial diagnosis is made and the person is referred to a specialty. In my setting, a patient outcome such as the PRM-Acute would be more meaningful than the LOS-ED.

I look forward to reading the results of the project in due course.

7. PLOS authors have the option to publish the peer review history of their article (what does this mean?). If published, this will include your full peer review and any attached files.

Reviewer #1: **Yes: **Simon Conroy

Reviewer #2: **Yes: **James van Oppen

---

## [Author Response · Author response to Decision Letter 0]

28 Jul 2023

Dear Mr. Bayer,

We thank you and the reviewers for your effort to assess our manuscript. The feedback on the manuscript was helpful and we feel that this helped us to improve the quality of our paper. We have edited the manuscript to address the concerns. All modified parts are marked with track changes. We have outlined the major changes and provided a point to point reply to the comments. 

The most important change of our article is renaming the study design as it should have been named. We agree with the reviewers that in we in fact did not meet the criteria for a stepped wedged design study. We also had to conclude that our aimed and described study design is a parallel cohort study with a baseline period. We have changed this in the title and abstract, and further elaboration on the design is added to the method section. We apologize for the inconvenience. 

Regarding the mentioned publication criteria error, two of the authors are indeed part of The ORCA Acute Medicine Research Consortium, however this proposed study is not initiated by this consortium. Therefore we did not mention the consortium and have not adjusted this in our manuscript, conform the applicable agreements within ORCA. 

Regarding data availability, this manuscript does not report any data and the data availability policy is not applicable to this article. 

We believe that the manuscript is now suitable for publication in PLOS One. 

Kind regards, on behalf of all authors,

M.G.A.M. van der Velde

 

Point to point response

Reviewer #1: 

Nice proposal and a much needed study. 

Dear reviewer, Mr. Conway,

We thank you for your effort to assess our manuscript. The feedback on the manuscript was helpful and we feel that this helped us to improve the quality of our paper. We have edited the manuscript to address the concerns.

Comment 1.1: I was unclear as to whether or not this was a cluster design, and if so why no randomisation, and if you needed to adjust the sample size for the cluster design. 

Reply 1.1: Thank you for your comment. We agree that we did not describe the study design properly and apologize for naming our study a stepped wedged design study while we do not meet criteria for this design. The aimed design of our study is a parallel cohort study with a baseline period. Due to the logistic restraints of implementing a care pathway in an acute care system that is under unprecedented strain, we did not randomize clusters. We started the intervention period conform convenience, sequentially following the capacity of participating hospitals to implement the pathway. We changed title, abstract and further elaboration on the design is given in the method section. We added a reference on parallel designs versus stepped wedged design. See page 1, line 3, page 2, line 25 , page 4, line 80 and the patient recruitment section on page 5-6.

[1] Hemming K, Haines T P, Chilton P J, Girling A J, Lilford R J. The stepped wedge cluster randomised trial: rationale, design, analysis, and reporting BMJ 2015; 350 :h391 doi:10.1136/bmj.h391

Comment 1.2: Also the intervention made no mention of other members of the team required to deliver CGA such as therapists. 

Reply 1.2: We agree that we did not mention the much needed help from allied health care professionals for performing a CGA and added this to the manuscript. See page 8, lines 166-169. 

Comment 1.3: Good to see a PROM in the outcomes, but perhaps also add in function?

Reply 1.3: We agree that evaluation of functional status would be of great value in this group of patients, but active measurements and follow-up of functional status was deemed not feasible in this study. Additionally, we do not expect the proposed care-pathway to primarily lead to improved functional outcomes. However, we did integrate the APOP-screener in the care-pathway. The APOP-screener expresses the risk of functional decline and we agreed that patients who are admitted and score high risk on the functional domain will receive consultation by a physical therapist during their hospital stay. Functional status is indeed an important outcome in this group of patients and for future studies we aim to implement active functional status measurements to evaluate this outcome. 

Reviewer #2: 

Thank you for inviting me to review this manuscript presenting a protocol for a stepped wedged cohort implementation study. Non-specific complaints are beginning to be recognised as the deadliest of presenting symptoms to geriatric emergency care. I might describe an NSC as a vaguely characterised symptom set outside of traditional teaching’s disease-based medical classification, which naively assumes that people can usually localise their problem. The classification presented in the methods is very helpful.

Dear reviewer, Mr. van Oppen,

We thank you and the reviewers for your effort to assess our manuscript. The feedback on the manuscript was helpful and we feel that this helped us to improve the quality of our paper. We have edited the manuscript to address the concerns.

Comment 2.1: For an international audience, I wonder if you might consider changing your first sentence from “older adults are frequently referred to the ED” to “older adults frequently attend the ED…” – outside of Belgium and the Netherlands people more commonly attend without referral, and NSC are no less serious in these cases.

Reply 2.1: Agreed. We changed it to older adults frequently attend the ED. See page 2, lines 19-21 and page 3, lines 42-43.

Comment 2.2: I notice the authors use a term “elderly care physicians”. I am interested to know how these professionals differ from geriatricians. In the UK, “elderly” has become an ageist slur and is a word best avoided if at all possible.

Reply 2.2: You make an excellent point about the avoidance of the word ‘elderly’, and we did avoid this term in our manuscript as much as possible. However, to our knowledge no other translation of ‘elderly care physician’ (ECP) exists. ECP’s are physicians who specializes in long-term care for older patients and work in nursing or residential homes and primary health care, and this function is unique in the world. The Dutch Association of Elderly Care Physicians is aware of the negative association with the word elderly and are currently looking for a better translation, until then the translation remains elderly care physician.

Comment 2.3: It is pleasing to see a patient-reported outcome being collected and that the researchers will include people with impaired cognition but who are able to answer the items. I wonder if you might consider adding a short statement acknowledging that capacity can fluctuate and that the participant will receive opportunity to re-attempt the PRM.

Reply 2.3: Although we think you make an excellent point about fluctuations in cognitive capacity, we are reluctant to add this statement. Although we feel reattempting the questionnaire could be of value, we can’t guarantee that all patients will have the opportunity to re-attempt due to logistic difficulties. 

Comment 2.4: Regarding the care pathway, I am not sure I understand why the diagnostics are being standardised. Does this imply that every person attending with a NSC will receive every stated test, including the controversial urinalysis? This approach would contradict my understanding of frailty risk and NSC as prompts for goal-oriented, person-centred care in which an investigation plan is tailored to the individual. I can see this ‘shotgun approach’ leading to confirmation bias in clinical decision-making. I also cannot see how analysis will differentiate the value of investigations, as for some NSCs a test e.g. chest xray will be appropriate and useful, whereas for others it will not add value.

Reply 2.4: We think you make an excellent point, this group of patients indeed need a goal-oriented, person-centered approach with a tailored investigation plan. However, treatment of older patients with NSC is challenging as shown by the high prevalence of adverse outcomes these patients experience. In our opinion these results show we are not yet in the place in which physicians can give these patients the best tailored care possible. The proposed set of diagnostic tests are considered standard care for the older ED-patient in a great number of Dutch ED’s. However, the advantages and possible disadvantages of standardized diagnostic screening and the different measurements is unknown. The secondary aim of our study is to evaluate standardization of diagnostic tests and the need for the different diagnostic measurements, including the controversial urine analysis. We believe this will help us guide a management protocol for patients with NSC and to make further recommendations, for example tailormade diagnostic trajectories following referral category. We added this in the objective, page 4, lines 76-77, and the discussion, page 11, lines 229-243.

Comment 2.5: The study is being powered to observe for reduction in LOS-ED. This for me prompts interesting reflection – in my practice, the LOS-ED is due to hospital capacity and crowding rather than patient factors, but presumably in all the research settings here the LOS-ED ends when an initial diagnosis is made and the person is referred to a specialty. In my setting, a patient outcome such as the PRM-Acute would be more meaningful than the LOS-ED. I look forward to reading the results of the project in due course.

Reply 2.5: Thank you for your comment. In the Netherlands the LOS-ED does not end when diagnosis is established and patients is referred to a specialty, it ends when the patient is discharged to the ward. LOS-ED is therefore associated with the number of consultations during the ED-visit, a higher number of diagnostic tests and experience of the evaluating physician. [1] By standardizing diagnostic tests and by agreeing experienced physicians will evaluate these patients, we expect to reduce the LOS-ED. However, we agree that LOS-ED is confounded by hospital capacity and crowding. We purposely did not power our study on patient reported measures, while we expect a considerable percentage of missing data due to the often high prevalence of cognitive disorders in the aimed population. However, in future projects it would be very meaningful to focus more on patient reported outcomes. We added this in the outcome measures, page 9, lines 188-196.

[1] Brouns SH, Stassen PM, Lambooij SL, Dieleman J, Vanderfeesten IT, Haak HR. Organisational Factors Induce Prolonged Emergency Department Length of Stay in Elderly Patients--A Retrospective Cohort Study. PLoS One. 2015 Aug 12;10(8):e0135066. doi: 10.1371/journal.pone.0135066. PMID: 26267794; PMCID: PMC4534295.

Reviewer 3: 

Dear reviewer, 

We thank you for your effort to assess our manuscript. The feedback on the manuscript was helpful and we feel that this helped us to improve the quality of our paper. We have edited the manuscript to address the concerns.

Comment 3.1: It is appreciated that the protocol suggests to use a novel research study design [The stepped wedge cluster randomised trial or cohort study which is increasingly being used in the evaluation of service delivery type interventions] with such mention in the title (A protocol for a multicenter stepped wedged cohort study), however, the actual design details are not described/given/clarified adequately. Even ‘Methods-Trial design’ section (lines 77-80, where details are expected) is very short and does not elaborate on this. It only says that “This study is a longitudinal multicenter cohort trial with a stepped wedged design. The study will be organized in hospitals in the Noord-Brabant area, the Netherlands, in which general practitioners (GPs), elderly care physicians (ECPs), Emergency Physicians (EPs), geriatricians and internists will collaborate. Participating hospitals can be general or teaching hospitals” which is not adequate. It is well recognised that design deserves as much consideration as analysis [This is pasted from one standard textbook on ‘Medical Research Methodology’] and I am sure that the authors already know this. According to my information/knowledge, this design involves random and sequential crossover of clusters from control to intervention until all clusters are exposed. A classical article on this design in BMJ by Hemming [Hemming K, Haines T P, Chilton P J, Girling A J, Lilford R J. The stepped wedge cluster randomised trial: rationale, design, analysis, and reporting BMJ 2015; 350 :h391 doi:10.1136/bmj.h391] is unfortunately (seems to had not used &) not quoted in the manuscript. This article says “In a stepped wedge study, the sample size calculation is complicated by the need to allow for the confounding effect of calendar time, and this means that the standard design effect is no longer applicable. Compared with a simple parallel study, where no such confounding occurs, the time effect tends to degrade the precision of the study and increase the sample size needed to achieve adequate power”. Even standard ‘design effect’ seems to have not been applied {or if applied is not mentioned/clarified in ‘Statistical Analysis-Sample size’ section (lines 179-191)}. If because of (due to) “crossover” cluster ‘design effect’ is not applicable, that needs to clarify and quote the reference. In fact, there is a complete absence of any reference on this design or methodology {I could not identify any such reference out 18 mentioned/listed in this manuscript}. Is not it surprising?

Reply 3.1: Thank you for your comment. We agree that we did not describe the study design properly and apologize for naming our study a stepped wedged design study while we do not meet criteria for this design. The aimed design of our study is a parallel cohort study with a baseline period. Due to the logistic restraints of implementing a care pathway in an acute care system that is under unprecedented strain, we did not randomize clusters. We started the intervention period conform convenience, sequentially following the capacity of participating hospitals to implement the pathway. We changed title, abstract and further elaboration on the design is given in the method section. We added the proposed reference on parallel designs versus stepped wedged design. See page 1, line 3, page 2, line 25 , page 4, line 80 and the patient recruitment section on page 5-6. 

[1] Hemming K, Haines T P, Chilton P J, Girling A J, Lilford R J. The stepped wedge cluster randomised trial: rationale, design, analysis, and reporting BMJ 2015; 350 :h391 doi:10.1136/bmj.h391

Comment 3.2 Is article quoted as number 17 [Kremers MNT, Mols EEM, Simons YAE, van Kuijk SMJ, Holleman F, Nanayakkara PWB, et al. Quality of acute internal medicine: A patient-centered approach. Validation and usage of the Patient Reported Measure-acute care in the Netherlands. PLOS ONE. 2020;15(12):e0242603] used for required sample size estimation? Since the design used and focus of that study is entirely different {In this study, we primarily aim to assess the validity of the PRM-acute care in internal medicine patients and secondly, to gain insight into the current perceived quality of acute care, with the overarching goal to use the PRM-acute care in daily practice and improve patient-centered care in the ED}, I wonder, ‘how’ that helped? Will you please explain? [essential since it is quoted here]. Because the earlier quoted Hemming’s article on page 4 says “As yet, there is no specific adaptation of design effects, for calculating the power or sample size in a cohort stepped wedge trial, nor implementation in a statistical package for this design.”, I would like to learn (as Hemming’s article is of 2015 & you may be aware or might have used later/newer developments).

Except these minor points, the article is acceptable. Nevertheless, mind you that as pointed out in ‘important note’ above “This review pertains only to ‘statistical aspects’ of the study and so ‘clinical aspects’ should be assessed separately/independently. ‘Minor Revision’ is recommended.

Reply 3.2: We apologize, the added reference is wrong. The data used for our power analysis is preliminary data from a prospective study about the quality of acute care, by the same author as the referred article, M.N.T. Kremers. These data are submitted and we adjusted the reference. See page 9, lines 199-203.

---

## [Decision Letter · Decision Letter 1]

13 Aug 2023

Implementation of a care-pathway at the emergency department for older people presenting with nonspecific complaints; A protocol for a multicenter parallel cohort study

PONE-D-23-17637R1

Dear Dr. van der Velde,

Thank you for your revised manuscript and for your considered attention to the reviewer comment. We’re pleased to inform you that your manuscript has been judged scientifically suitable for publication and will be formally accepted for publication once it meets all outstanding technical requirements.

Kind regards,

Antony Bayer

Academic Editor

PLOS ONE

Additional Editor Comments (optional):

Reviewers' comments:

Reviewer's Responses to Questions

**Comments to the Author**

1. Does the manuscript provide a valid rationale for the proposed study, with clearly identified and justified research questions?

Reviewer #2: Yes

Reviewer #3: Yes

2. Is the protocol technically sound and planned in a manner that will lead to a meaningful outcome and allow testing the stated hypotheses?

Reviewer #2: Yes

Reviewer #3: Yes

3. Is the methodology feasible and described in sufficient detail to allow the work to be replicable?

Reviewer #2: Yes

Reviewer #3: Yes

4. Have the authors described where all data underlying the findings will be made available when the study is complete?

Reviewer #2: Yes

Reviewer #3: Yes

5. Is the manuscript presented in an intelligible fashion and written in standard English?

Reviewer #2: Yes

Reviewer #3: Yes

6. Review Comments to the Author

You may also provide optional suggestions and comments to authors that they might find helpful in planning their study.

Reviewer #2: My concerns have been addressed and I thank the authors for their attentiveness to my comments. Good luck with the study.

Reviewer #3: COMMENTS: All the comments are answered and positively attended [good that the study title is changed]. I recommend the acceptance because the manuscript has now achieved the acceptable level in my opinion.

7. PLOS authors have the option to publish the peer review history of their article (what does this mean?). If published, this will include your full peer review and any attached files.

Reviewer #2: No

Reviewer #3: **Yes: **Dr. Sanjeev Sarmukaddam

---

## [Editor Report · Acceptance letter]

21 Aug 2023

PONE-D-23-17637R1 

Implementation of a care-pathway at the emergency department for older people presenting with nonspecific complaints; A protocol for a multicenter parallel cohort study 

Dear Dr. van der Velde:

I'm pleased to inform you that your manuscript has been deemed suitable for publication in PLOS ONE. Congratulations! Your manuscript is now with our production department. 

Kind regards, 

on behalf of

Professor Antony Bayer 

Academic Editor

PLOS ONE